# Pro-SMP finder–A systematic approach for discovering small membrane proteins in prokaryotes

Tara Hoffman[1], Jeff Kinne[1]*, Kyu Hong Cho[2]*

1 Department of Math and Computer Science, Indiana State University, Terre Haute, Indiana, United States of America, 2 Department of Biology, Indiana State University, Terre Haute, Indiana, United States of America

* jkinne@indstate.edu (JK); kyuhong.cho@indstate.edu (KHC)

**Data Availability Statement:** All files are available from http://cs.indstate.edu/pro-smp-finder.

**Funding:** NIH 5R25MD011712-04 The funders had no role in study design, data collection and analysis, decision to publish, or preparation of the manuscript.

## Abstract

Prokaryotic chromosomes contain numerous small open reading frames (ORFs) of less than 200 bases. Since high-throughput proteomics methods often miss proteins containing fewer than 60 amino acids, it is difficult to decern if they encode proteins. Recent studies have revealed that many small proteins are membrane proteins with a single membrane-anchoring α-helix. As membrane anchoring or transmembrane motifs are accurately identifiable with high confidence using computational algorithms like Phobius and TMHMM, small membrane proteins (SMPS) can be predicted with high accuracy. This study employed a systematic approach, utilizing well-verified algorithms such as Orfipy, Phobius, and Blast to identify SMPs in prokaryotic organisms. Our main search parameters targeted candidate SMPs with an open reading frame between 60–180 nucleotides, a membrane-anchoring or transmembrane region 15 and 30 amino acids long, and sequence conservation among other microorganisms. Our findings indicate that each prokaryote possesses many SMPs, with some identified in the intergenic regions of currently annotated chromosomes. More extensively studied microorganisms, such as *Escherichia coli* and *Bacillus subtilis*, have more SMPs identified in their genomes compared to less studied microorganisms, suggesting the possibility of undiscovered SMPs in less studied microorganisms. In this study, we describe the common SMPs identified across various microorganisms and explore their biological roles. We have also developed a software pipeline and an accompanying online interface for discovering SMPs (http://cs.indstate.edu/pro-smp-finder). This resource aims to assist researchers in identifying new SMPs encoded in microbial genomes of interest.

## Introduction

Recent advances in high-throughput DNA sequencing technologies have enabled the decoding of genetic information in various organisms, leading to the discovery of many new open-reading frames (ORFs). Unfortunately, many small ORFs encoding less than 60 amino acids have been overlooked due to their size being smaller than the minimal ORF length cutoff typically

**Competing interests:** The authors have declared that no competing interests exist.

used during gene annotations. Consequently, a substantial number of small proteins have yet to be discovered. In *E. coli*, small proteins have been extensively searched experimentally, and it has been found that a third of are membrane-anchored or transmembrane proteins with an α-helical transmembrane motif [1, 2]. Since α-helical membrane-anchoring or transmembrane motifs can be predicted with bioinformatics tools such as Phobius [3–5] and TMHMM [6, 7] with high accuracy, we searched for potential small membrane proteins (SMPs) using these bioinformatic tools.

Many SMPs have been identified as versatile regulators at the cellular membrane, engaging in interactions with membrane proteins and participating in a wide range of cellular processes. The processes include transport, signal transduction, stress response, respiration, cell division, sporulation, and membrane stability. Predominantly, SMPs are associated with the membrane using an α-helical conformation [8, 9]. These α-helix domains in the membrane appear to facilitate interaction with other membrane proteins or sense membrane physiological conditions.

SMPs have been identified in nearly all organisms, including eukaryotes, where they also exhibit diverse biological functions. Unlike prokaryotes, however, the presence of introns in eukaryotic genes complicates the accurate prediction of stand-alone SMPs, as they may be part of an integral membrane protein with multiple transmembrane motifs. Consequently, this study has excluded eukaryotic SMPs from its scope.

## Methods

A major aim of this study is to create a pipeline to search for potential SMPs that can be applied to any prokaryotic organism, is fast enough to run as a systematic search on many genomes, and is as simple as possible while still accomplishing our goals. We combined existing well-verified tools that are flexible and efficient into a pipeline.

### The overview of the pipeline, Pro-SMP

Our pipeline contains the following main steps. (1) Download of genome assembly and annotation. (2) Identification of already annotated coding sequences and intergenic open reading frames (ORFs) that are of the desired length. (3) Predicting protein topology if candidate ORFs code for transmembrane portion. (4) Estimating whether candidates are conserved within related species. (5) Checking whether candidates are expressed in a given RNA expression dataset (when this option is enabled). Starting from step (2), we have a list of candidate ORFs and genes, and each subsequent step narrows the list of candidates based on settings that the user can change. The final list of candidates is annotated with the results of each of the steps in the pipeline. See **Fig 1** for a graphic of the steps involved in the pipeline. The software repository for the source code for the pipeline also contains scripts that can be used to batch execute the pipeline on a large number of species.

**Genome download.** The pipeline starts by downloading a genome assembly and annotation. The user supplies an NCBI genome assembly id. For example, the ID GCF_000005845.2_ASM584v2 is the reference assembly for the K-12 strain of *E. coli*. The pipeline downloads the full genome and list of annotated genes. An assembly ID can be easily looked up using NCBI's online taxonomy browser or the KEGG online database; we also include a script for looking up assembly IDs for a list of species. We have run our pipeline on dozens of prokaryotic organisms, and the assembly IDs we used are included in both **S2 Table in S1 File** and as part of the software's source code repository.

**Identification of coding sequences and intergenic ORFs.** The user can set options for the desired length of candidate genes (default is to search for proteins with 20–60 amino acids)

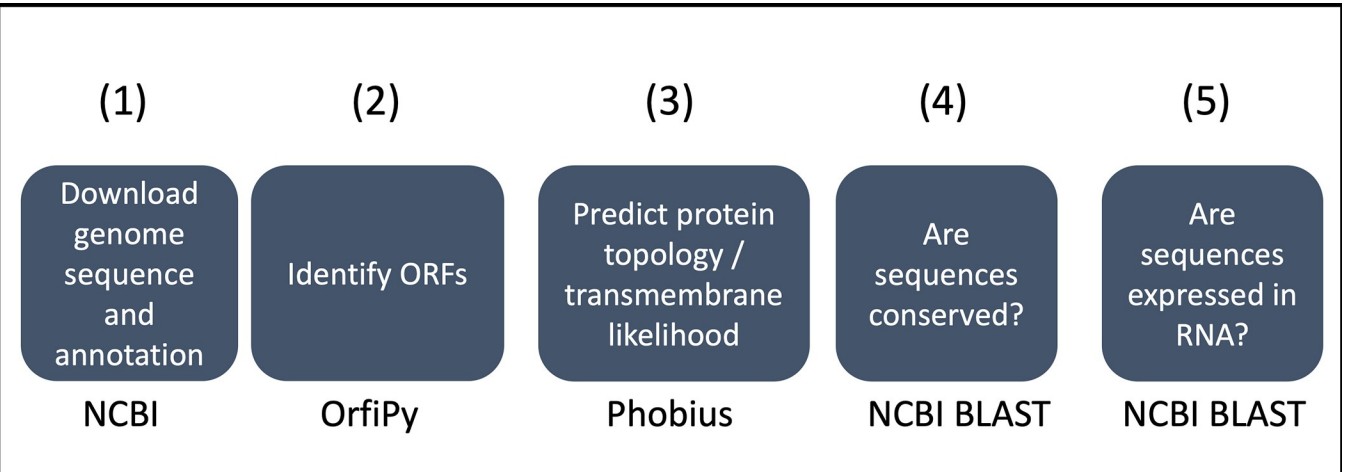

**Fig 1. The main stages of the pipeline are indicated in the boxes.** The text under each box indicates the main software or tool that is utilized in that stage. Python scripts tie the stages together, provide a user interface, allow for configuration of options, and output results in a variety of formats. The web interface of the software runs the pipeline from a server, allowing the pipeline to be run with no software installation required for the user.

and start/stop codons (default is the standard set for bacteria). The pipeline extracts annotated coding sequences that match the search parameters and extracts potential novel ORFs from intergenic regions using the Python tool Orfipy [10]. Orfipy was selected to use for identifying open reading frames due to its relative simplicity, efficiency, and ability to download and run locally. Other popular ORF detection tools could have been used (e.g., ORF Finder [11]).

**Predicting protein topology.** Each candidate sequence that has passed previous steps of the pipeline has its topology estimated using the Perl tool Phobius [3–5]. Phobius produces an estimate of the likelihood that the protein sequence would contain any transmembrane portions and whether it is likely to be a protein with a secretion signal sequence. User-configurable options (how many transmembrane regions a candidate can have, range of lengths for transmembrane regions, retain or remove candidates that Phobius labels as signal peptides) are used to determine which candidates are retained from this stage of the pipeline. Our default parameters retain candidates that are predicted to have a transmembrane portion of the peptide that is between 15 and 30 amino acids long and include signal peptides.

**Conservation of candidates.** Candidate sequences are blasted using a tblastn search for hits of the candidates among prokaryotic genomes that are *not* within a specified taxonomy relationship. Our results are based on searching for blast hits against genomes that are not within the same species. If there are blast hits outside of the same species, it is likely that the sequence is conserved and, therefore, also for coding a protein. Our default parameters set a threshold of 30 for blast hits to retain a candidate. We note that this setting, along with many others, can be adjusted depending on the desired sensitivity of the search.

**RNA expression.** If the user supplies an SRA ID of an RNA-seq dataset (e.g., ID SRX4985301 [12] is an RNA-seq dataset for wild-type samples of *E. coli.*), the pipeline uses NCBI tblastn to blast the candidate sequences against the RNA-seq dataset to determine for each if the candidate is likely being expressed in that sample. A user-configurable cutoff is used as a threshold to determine which candidates to retain. Our default settings skip this step, but if the step is enabled, the default cutoff is set to 30 for the number of hits required to retain a candidate.

**User interface and results.** The pipeline can be downloaded from the project's Gitlab repository [13] and run locally, or the pipeline can be invoked from a web interface to an installation of the pipeline (so that no software installation and configuration is necessary). A

zip file is available for download that contains the results of each stage of the pipeline. The final results contain the following for each candidate: identifier, nucleotide and peptide sequence, location in genome, predicted topology from Phobius, number of blast hits to other species microorganisms, number of blast hits to RNA-seq dataset, and summary of results.

The local interface runs from the terminal and includes information on installation and configuration. The Gitlab repository documents prerequisites for the pipeline, which includes Phobius, Perl, Python3, certain Python modules (Orfipy, Pyfastx, BioPython, GFFutils), NCBI blast, and edirect locally installed. The pipeline can run within Windows, Mac OS, or Linux.

## Pro-SMP–Details on choices for parameters and tools

There are a wide variety of tools available that predict whether a given peptide sequence is likely to have a transmembrane moiety or not. We selected Phobius because it is efficient enough to run quickly on thousands of candidate genes, can be downloaded and run locally, and has good accuracy. Many protein topology tools are either much more computationally expensive (e.g., AlphaFold [14, 15], Deep-TMHMM [16], TOPCONNS2 [17]), are not optimized to work as well for the short length of proteins we consider (GLIMMER3 [18], Prodigal [19], GeneMark [20–22]), or would have similar performance in our pipeline as Phobius (e.g., TMHMM [6, 7]) based on our testing. Full annotation pipelines also could be used (e.g., PGAP [23–25], RASTtk [26–28], Prokka [29]) but are more computationally expensive and normally rely on only previous annotated genes (not optimized to search for intergenic small orfs). Because we are most interested in very short proteins with an α-helix transmembrane moiety, Phobius is well-suited for our purposes. We note that Phobius is limited to predicting proteins with alpha helices. Our normal range of parameters would be too short for a transmembrane beta-sheet.

During our search, we constrained protein size, limiting it to 20 to 60 amino acids. It appears that the majority of SMPs of *E. coli* contain an α-helix in the α-fold analysis in the UniProt database (Unpublished Data). Proteins' α-helix consists of 3.6 amino acid residues per turn, with each turn stretching 5.4 Angstrom. Given that the typical thickness of cytoplasmic membranes including the phosphate head group is around 4 nanometers (40 Angstroms), an α-helix would require roughly 7.4 turns to penetrate the membrane, which translates to roughly 26.6 amino acids. When we analyzed transmembrane proteins using Phobius, it appeared that 21–22 amino acid α-helix were enough to traverse the membrane, likely due to the head group of phospholipids at both sides of the membrane. With our default value (15 to 30 amino acids), both small proteins penetrating the membrane and partially inserted into the membrane can be found.

For the search for SMPs using Phobius, we included signal sequences because some candidates have signal sequence characteristics even though they are SMPs. (e.g., prophage toxic proteins HokC and HokD, PhoQ kinase inhibitor MgrB, MntS, $K^+$ transporting P-type ATPase subunit KdpF, and cytochrome bd-I ubiquinol oxidase accessory subunit CydX).

To identify homologs of SMP candidates in our pipeline, we utilized NCBI Blast with default parameter settings. We excluded the same species of the microorganism analyzed in the search due to the sequence similarity between strains within a species. This would increase the probability of discovering genuine SMPs in intergenic regions. With the default setting of more than or equal to 30 homologs, 41 putative SMPs among the annotated genes in *E. coli* (GCF_000005845.2_ASM584v2) were identified (**S2 Table in S1 File**). The default number of homologs can be adjusted to suit specific research needs.

The utilization of RNAseq data to examine the expression of SMP candidate genes presents a valuable approach, particularly for those located in intergenic regions. Since intergenic

regions were typically excluded from gene annotation, the detection of RNA expression from intergenic regions could indicate that candidate genes located in these regions may be genuine. However, it should be noted that genes may only be expressed under specific conditions, so the absence of gene expression in certain conditions does not necessarily imply their non-expression in other conditions. It should also be noted that, at times, intergenic regions can be expressed as part of an RNA transcript from neighboring genes. Thus, it is important to carefully monitor the expression profile of candidates in RNAseq data, as well as the presence of translation signatures, such as ribosome binding sites preceding candidate genes, to confirm their independent expression. Our default setting for the inclusion of RNAseq data is currently inactive. When activated, the default threshold for the number of hits required to retain a candidate is set at 30, and this value is customizable by users.

## Results and discussion

We have developed a pipeline that combines several existing bioinformatic tools to automatically find candidate SMPs for a given organism with parameters tuned for this task. The pipeline is available for download or can be utilized through a web interface. The main stages of our pipeline are–download and processing of genome assembly, identification of coding sequences and potential additional ORFs within intergenic regions, determination of transmembrane likelihood, determination of conservation among different species, determination of expression in wet-lab RNA-seq sample. Each stage filters the initial candidate gene and ORF list with configurable parameters, and at the end we have a candidate list of potential SMPs for further consideration. The final list includes information gleaned from the bioinformatic tools at each stage. Many parameters can be tuned for the entire process, which are all easily configurable.

Each stage of our pipeline plays a key role in generating the candidate list and filtering out unlikely candidates. We begin with a large candidate list of annotated coding sequences and ORFs from intergenic regions. This initial list is much too large (thousands of candidates, or more) to carefully consider each candidate. The initial list is filtered to ensure each final candidate has the following properties: is predicted to have a transmembrane or membrane-associated structure, is preserved across different species, and has evidence of expression from RNA-seq data. Candidates passing each of these tests are worthy of further close study.

Some SMPs have the pattern of a signal sequence, which typically comprises one or more positively charged amino acid residues at the N-terminus followed by a stretch of 6–12 hydrophobic residues and a cleavage site (e.g., prophage toxic proteins HokC and HokD, PhoQ kinase inhibitor MgrB, MntS, K$^+$ transporting P-type ATPase subunit KdpF, and cytochrome bd-I ubiquinol oxidase accessory subunit CydX). This might indicate that some SMPs might have originated from signal sequences of other proteins. The length of the *h*-region (hydrophobic region) in signal sequences is typically 5 to 16 amino acids [3], which is insufficient to traverse the membrane, so these would have become SMPs partially inserted into the membrane.

When signal sequences were included in the Phobius search, the *E. coli* leucine leader peptide was found as an SMP. The leucine leader peptide is a short sequence of amino acids encoded within the messenger RNA (mRNA) of genes involved in the biosynthesis of branched-chain amino acids (BCAAs). It regulates the production of enzymes synthesizing BCAAs through the mechanism called attenuation. The leader peptide forms a perfect α-helix transmembrane structure in α-fold, suggesting that this leader peptide might become an SMP when it is terminated in front of the downstream genes and have another function as an SMP in the membrane.

To confirm the accuracy of our pipeline, we have performed a detailed analysis of our pipeline's predictions for the well-studied organism *E. coli*. With optimal settings, our pipeline has a false positive rate of 5.8% and a false negative rate of 2% for the testing set we identified. The testing set is given in **S1 Table in S1 File**. We have also run our pipeline on dozens of prokaryotic organisms to see the degree to which there exist potential undiscovered SMPs across different species. We have found that each organism tested has at least a few potential unstudied SMPs in both coding and noncoding (intergenic) sequences. They are listed in the **S2 and S3 Tables in S1 File**. These commonly identified SMPs are different between Gram-positive and Gram-negative bacteria. The common SMPs are described here.

## SMPs commonly identified in Gram-positive bacteria

- SecE: all bacteria possess a general protein secretion system, the Sec system, that is responsible for the transport of most extracellular proteins. The Sec system consists of two main components, SecA, an ATPase providing energy for the protein translocation, and SecYEG, a heterotrimeric transmembrane channel complex [30]. SecY consists of 10 α-helical transmembrane segments and forms the actual channel. SecE of Gram-positive bacteria are SMPs. It is composed of mostly 50 to 70 amino acids but some bacteria such as *Cruptobacterium curtum* and *Raineyella fluvialis* have SecE with more than 140 amino acids. Regardless of lengths, all have a single α-helical transmembrane motif, but little amino acid homology is found among the membrane-spanning segments. Instead, most conservation is found in their cytoplasmic amino termini, suggesting that the membrane-spanning segment has a structural role and the cytoplasmic segment has the functional role [31]. Gram-negative version of SecE, for example, *E. coli* SecE, has three transmembrane helices, so Gram-negative SecE proteins are larger than Gram-positive counterparts [32]. Even though Gram-positive SecE is small, it is essential for protein translocation, as with *E. coli* SecE [33].

- TatAy: The twin-arginine protein translocation (Tat) system translocates folded and cofactor-containing proteins across the membrane [34]. To be recognized by the Tat system, folded proteins should have twin arginine residues in the signal sequence. The Tat system is found in the membrane of many archaea and bacteria. The system comprises two main complexes, a docking complex and a pore complex. The docking complex is composed of TatC and a TatA or TatA-like protein. TatC has six membrane-spanning domains. The pore complex is formed by TatA molecules, each of which has a transmembrane domain. In *B. subtilis*, there are two Tat systems, TatAyCy and TatAdCd. TatAy and TatAd form pores and interact with TatCy or TatCd, respectively [35]. Among those proteins, TatAy is an SMP.

- DltX: The Dlt system in Gram-positive bacteria incorporates D-alanine to teichoic acids on the cell surface. D-alanine incorporation reduces the negative charges of the teichoic acids, so prevents cationic antimicrobial peptides (CAMPs) from accessing bacterial cell surface targets such as cell walls or cell membranes. In most Gram-positive bacteria, the *dlt* operon is composed of five genes *dltXABCD*. DltX is an SMP and essential for D-alanine incorporation [36]. The exact role of DltX in the process of D-alanylation of teichoic acids, however, is not known.

- Fst family type I toxin-antitoxin system toxins: The Fst family type I toxin-antitoxin system is ubiquitous in the genome of Firmicutes (Gram-positive bacteria with a low G+C content) [37]. Our search found that the toxins of the system in many Gram-positive bacteria are SMPs. The toxin molecules of the type I toxin-antitoxin system are peptides and the antitoxin molecules are RNAs that interfere with the translation of mRNA encoding its

corresponding toxin [38]. The overexpression of these toxins in bacteria lyses bacterial membranes [39], but it is not known whether or not these toxins directly lyse the bacterial membrane. These toxins encoded in plasmids act as addiction molecules for plasmid maintenance [40]. However, the role of the chromosomally encoded Fst family type I toxin-antitoxin system has not been elucidated yet [41].

- Phage holin-like toxin: Holins are one of the essential proteins for bacteriophages to lyse host cells. Holins are small membrane proteins that permeabilize endolysins, which are phage enzymes accumulated in the host cytosol and degrade host bacterial cell walls. Holins have one to four transmembrane spanning α-helices [42] and oligomerize to form pores or channels in the membrane [43]. The holins with one transmembrane α-helix are SMPs.

- Photosystem-involved membrane proteins in photosynthetic bacteria. Bacterial photosystems are composed of three membrane protein complexes, photosystem I, photosystem II, and cytochrome b6/f complex. Each complex is composed of many integral membrane proteins. In our search, some proteins in the complex are SMPs with one transmembrane α-helix. They are photosystem I reaction center subunit VIII (PsaI), IX (PsaJ), photosystem II reaction center protein J (PsbJ), N (PsbN), and X (PsbX), photosystem II cytochrome b559 subunit β, and cytochrome b6/f complex subunit M (PetM) in *Synechococcus elongatus*. In another photosynthetic bacterium *Prochlorococcus marinus*, the photosystem I reaction center subunit VIII (PsaI), IX (PsaJ), photosystem II reaction center protein I (PsbI), K (PsbK), L (PsbL), M (PsbM), N (PsbN), T(PsbT) and cytochrome b559 subunit β, and cytochrome b6/f complex subunit 5 (PetG) are SMPs. Thus, not all SMPs in a photosynthetic bacterium are SMPs in another bacterium but there is quite a bit of overlap between bacteria.

## SMPs commonly identified in Gram-negative bacteria

- SecE: SecE in many Gram-negative bacteria are also found as SMPs.

- AcrZ: Multidrug efflux pumps are used by many microorganisms to overcome the toxicity of antimicrobial agents. One of the Gram-negative multidrug efflux pumps, the AcrAB-TolC multidrug efflux pump can pump out diverse antibiotics [44]. The membrane protein AcrB that makes an inner membrane pore interacts with AcrZ, an SMP with 49 amino acids in *E. coli*. A previous study indicates that AcrZ may enhance the export ability of a subgroup of antimicrobial agents by modulating the AcrB activity [45].

- CcmD: CcmD is a component of the CcmABC ATP-binding cassette transporter complex that delivers heme from cytoplasm to periplasm for cytochrome C maturation in Gram-negative bacteria. CcmD is an SMP with the topology of an N-terminus outside and C-terminus inside [46].

- CydX, CcoQ, and CcoS: Cytochrome complexes are responsible for electron flow in oxidative phosphorylation reactions. Since this electron flow occurs in or on the membrane, most cytochrome complex subunits are membrane proteins, and some cytoplasmic complex subunits such as CydX, CcoQ, and CcoS are SMPs. CydX is a subunit of Cytochrome bd-1 ubiquinol oxidase complex, which transfers 4 electrons from ubiquinol to oxygen and produces 4 protons in the periplasmic space and 2 water molecules [47]. Since cytochrome bd oxidase exists only in prokaryotes, it is considered a plausible drug target. Cytochrome C oxidase is found in the mitochondria of eukaryotes and aerobic prokaryotes. It is a large transmembrane protein complex and the last enzyme in the respiratory electron transport chain. Some of the subunits of cbb 3-type such as CcoQ and CcoS are SMPs in many bacteria.

### SMPs identified in archaea

- Archaea also possess SMPs, but conserved ones are not as many as those in bacteria. Most conserved SMPs are currently listed as hypothetical proteins since they have not been studied yet. Two of the conserved SMPs are Sec61β and Sec61γ, components of the protein secretion system Sec61. This protein transport system is an ortholog of SecYEG in bacteria. The Sec61 transport system is composed of three membrane proteins: Sec61α, Sec61β, and Sec61γ. Sec61β Sec61γ are orthologs of SecG and SecE, respectively. Eukaryotes also possess the Sec61 system, which is the central component of the protein translocation apparatus of the endoplasmic reticulum (ER) membrane.

### Putative SMPS identified from intergenic sequences

Our search pipeline has identified numerous potential SMPs within the intergenic sequences of nearly all microbial genomes (**S2 Table in S1 File**). This suggests that many organisms likely produce a significant number of SMPs that are yet to be discovered.

## Conclusion

Small membrane-associated proteins (SMPs) are proteins that are involved in important cellular processes such as transport, signal transduction, stress response, respiration, cell division, sporulation, and membrane stability. They are inserted into the membrane through mostly α-helical conformation and interact with other membrane proteins or sense membrane physiological conditions. Our search pipeline identified more SMPs from designated ORFs of well-studied microbes such as *E. coli* (41 SMPs), *Klebsiella pneumoniae* (40), *Salmonella enterica* (30), *Vibrio cholerae* (27), and *B. subtilis* (26 SMPs) than those from less studied microorganisms (mostly less than 10 SMPs) (**S2 Table in S1 File**). This result suggests that many SMPs in less studied microorganisms are not yet discovered and may be encoded in intergenic regions. Further research and characterization of these small transmembrane proteins can help elucidate their biological roles and contribute to a better understanding of prokaryotic genomes. The software pipeline and online interface used in this study can be a valuable resource for researchers who investigate microbial SMPs.

## Supporting information

**S1 File. Supplementary data.**
(DOCX)

## Author Contributions

**Conceptualization:** Kyu Hong Cho.

**Data curation:** Tara Hoffman, Jeff Kinne.

**Formal analysis:** Tara Hoffman, Jeff Kinne, Kyu Hong Cho.

**Funding acquisition:** Jeff Kinne.

**Investigation:** Tara Hoffman, Jeff Kinne, Kyu Hong Cho.

**Methodology:** Tara Hoffman, Jeff Kinne, Kyu Hong Cho.

**Project administration:** Jeff Kinne, Kyu Hong Cho.

**Resources:** Jeff Kinne.

**Software:** Tara Hoffman, Jeff Kinne.

**Supervision:** Jeff Kinne, Kyu Hong Cho.

**Validation:** Tara Hoffman, Jeff Kinne, Kyu Hong Cho.

**Visualization:** Tara Hoffman, Jeff Kinne, Kyu Hong Cho.

**Writing – original draft:** Jeff Kinne, Kyu Hong Cho.

**Writing – review & editing:** Jeff Kinne, Kyu Hong Cho.

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
