## [Decision Letter · Decision Letter 0]

18 Dec 2023

PONE-D-23-37357Pro-SMP finder–a systematic approach for discovering small membrane proteins in prokaryotesPLOS ONE

Dear Dr. Cho,

Thank you for submitting your manuscript to PLOS ONE. After careful consideration, we feel that it has merit but does not fully meet PLOS ONE’s publication criteria as it currently stands. Therefore, we invite you to submit a revised version of the manuscript that addresses the points raised during the review process.

We look forward to receiving your revised manuscript.

Kind regards,

Amitava Mukherjee, ME, Ph.D.

Academic Editor

PLOS ONE

Journal Requirements:

Whilst you may use any professional scientific editing service of your choice, PLOS has partnered with both American Journal Experts (AJE) and Editage to provide discounted services to PLOS authors. Both organizations have experience helping authors meet PLOS guidelines and can provide language editing, translation, manuscript formatting, and figure formatting to ensure your manuscript meets our submission guidelines. To take advantage of our partnership with AJE, visit the AJE website (http://aje.com/go/plos) for a 15% discount off AJE services. To take advantage of our partnership with Editage, visit the Editage website (www.editage.com) and enter referral code PLOSEDIT for a 15% discount off Editage services. If the PLOS editorial team finds any language issues in text that either AJE or Editage has edited, the service provider will re-edit the text for free.

NIH 5R25MD011712-04

5. We notice that your supplementary tables are included in the manuscript file. Please remove them and upload them with the file type 'Supporting Information'. Please ensure that each Supporting Information file has a legend listed in the manuscript after the references list.

Reviewers' comments:

Reviewer's Responses to Questions

**Comments to the Author**

1. Is the manuscript technically sound, and do the data support the conclusions?

Reviewer #1: Yes

2. Has the statistical analysis been performed appropriately and rigorously? 

Reviewer #1: Yes

3. Have the authors made all data underlying the findings in their manuscript fully available?

Reviewer #1: Yes

4. Is the manuscript presented in an intelligible fashion and written in standard English?

Reviewer #1: Yes

5. Review Comments to the Author

Reviewer #1: This study employed a systematic approach, utilizing well-verified algorithms such as Orfipy, Phobius, and Blast to identify SMPs in prokaryotic organisms. Their findings indicated that each prokaryote possesses many SMPs, with some identified in the intergenic regions of currently annotated chromosomes. In this study, they have developed a software pipeline and an online interface for discovering SMPs (http://cs.indstate.edu/pro-smp-finder). This resource assists researchers in identifying new SMPs encoded in microbial genomes of interest. I have only one important concern. The authors should improve the pipeline and online interface of the Pro-SMP Finder software to enhance its usability and broaden its applications.

6. PLOS authors have the option to publish the peer review history of their article (what does this mean?). If published, this will include your full peer review and any attached files.

Reviewer #1: No

---

## [Author Response · Author response to Decision Letter 0]

1 Feb 2024

Response to the editorial and reviewer’s comments.

• Done.

2. We notice that your supplementary tables are included in the manuscript file. Please remove them and upload them with the file type 'Supporting Information'. Please ensure that each Supporting Information file has a legend listed in the manuscript after the references list. 

• Done.

3. The authors should improve the pipeline and online interface of the Pro-SMP Finder software to enhance its usability and broaden its applications.

• The items below are improved to enhance the usability and broaden its application.

o The web interface to the pipeline has been streamlined and improved in the following ways.

o Results can now be viewed in the web browser after the pipeline completes. Results will remain on the server for at least 2 weeks, and a download link is provided to download the results locally.

o Results now include links to gene pages at NCBI and UniProt to facilitate further exploration.

o Results now include an improved summary of the running of the pipeline and results from each intermediate step of the pipeline (how many genes passed each stage, how to determine where a gene was filtered out).

o Results now include a genome browser view so that each candidate gene or intergenic region can be quickly seen in the context of other genes.

o The listing of full results has been updated to be easier to understand.

o The interface for the web submission form has been improved to be more intuitive and easy to use.

---

## [Decision Letter · Decision Letter 1]

6 Feb 2024

Pro-SMP finder–a systematic approach for discovering small membrane proteins in prokaryotes

PONE-D-23-37357R1

Dear Dr. Cho,

We’re pleased to inform you that your manuscript has been judged scientifically suitable for publication and will be formally accepted for publication once it meets all outstanding technical requirements.

Kind regards,

Amitava Mukherjee, ME, Ph.D.

Academic Editor

PLOS ONE

Additional Editor Comments (optional):

Reviewers' comments:

Reviewer's Responses to Questions

**Comments to the Author**

1. If the authors have adequately addressed your comments raised in a previous round of review and you feel that this manuscript is now acceptable for publication, you may indicate that here to bypass the “Comments to the Author” section, enter your conflict of interest statement in the “Confidential to Editor” section, and submit your "Accept" recommendation.

Reviewer #1: All comments have been addressed

2. Is the manuscript technically sound, and do the data support the conclusions?

Reviewer #1: Yes

3. Has the statistical analysis been performed appropriately and rigorously? 

Reviewer #1: Yes

4. Have the authors made all data underlying the findings in their manuscript fully available?

Reviewer #1: Yes

5. Is the manuscript presented in an intelligible fashion and written in standard English?

Reviewer #1: Yes

6. Review Comments to the Author

Reviewer #1: (No Response)

7. PLOS authors have the option to publish the peer review history of their article (what does this mean?). If published, this will include your full peer review and any attached files.

Reviewer #1: No

---

## [Editor Report · Acceptance letter]

21 Feb 2024

PONE-D-23-37357R1 

PLOS ONE

Dear Dr. Cho, 

I'm pleased to inform you that your manuscript has been deemed suitable for publication in PLOS ONE. Congratulations! Your manuscript is now being handed over to our production team.

Kind regards, 

on behalf of

Professor Dr. Amitava Mukherjee 

Academic Editor

PLOS ONE